# Effects of Self-Talk Training on Competitive Anxiety, Self-Efficacy, Volitional Skills, and Performance: An Intervention Study with Junior Sub-Elite Athletes

**DOI:** 10.3390/sports7060148

**Published:** 2019-06-19

**Authors:** Nadja Walter, Lucie Nikoleizig, Dorothee Alfermann

**Affiliations:** 1Faculty of Sport Science, Institute of Sport Psychology and Physical Education, Leipzig University, 04109 Leipzig, Germany; alfermann@uni-leipzig.de; 2Faculty of Life Sciences, Institute of Personality Psychology and Psychological Assessment, Leipzig University, 04109 Leipzig, Germany; lucie.nikoleizig@uni-leipzig.de

**Keywords:** junior elite athletes, competitive anxiety, psychological skills training, self-efficacy, self-talk

## Abstract

(1) Background: Self-talk (ST) is used to influence athletes’ thoughts, feelings, and behaviors. Samples of squad and competitive athletes are underrepresented, although research has proven the positive effects of ST in the context of sports. Thus, the present study focused on the impact of ST on psychological and performance outcomes of junior sub-elite athletes. (2) Methods: *N* = 117 athletes (55 females, 62 males; *M* = 16.0 years) were randomly assigned to either one of two experimental groups or to a control group (*n* = 30). The experimental groups received an ST intervention for either one week (*n* = 36) or eight weeks (*n* = 38), and the control group received no ST training. The dependent variables (competitive anxiety, volitional skills, self-efficacy, and coaches’ performance ratings) were assessed three times before and after the intervention. It was expected that (a) an ST intervention would reduce the competitive anxiety and increase volitional skills, self-efficacy, and performance; and, (b) long-term training would lead to higher effects than short-term training. (3) Results: As expected, ST training led to (less) somatic state anxiety and (higher) state self-confidence, self-optimization, self-efficacy, and performance. Additionally, long-term training was more effective than short-term training. (4) Conclusions: Targeted ST interventions may help to improve junior athletes’ psychological states and performance.

## 1. Introduction

Self-talk (ST) is a cognitive technique that is used by athletes as the result of, or as a means of, influencing thoughts, feelings, and behaviors [1]. As such, ST “refers to those automatic statements that are reflective of, and deliberate techniques (e.g., thought-stopping) athletes use to direct sports-related thinking” [1] (p. 38). Van Raalte, Vincent, and Brewer [2] distinguish between automatic, spontaneous, and often emotionally driven ST on the one hand and directed, consciously driven ST on the other. The authors compare the former type of ST to information processing System 1 and the latter type to System 2 based on dual-process theories of cognitive psychology. Each system serves different functions: System 1 ST “represents the immediate, emotionally charged reaction to a situation” [2] (p. 140), whereas “System 2 self-talk results from consideration and planning” (p. 140). In a similar vein, Latinjak, Zourbanos, López-Ros, and Hatzigeorgiadis [3], as well as Latinjak, Hatzigeorgiadis, and Zourbanos [4], distinguish between undirected or spontaneous ST and goal-directed ST. These authors corroborate the assumptions of the dual-process model in their finding that both types of ST serve different functions, specifically, evaluative and time-perspective in the case of undirected ST, and activation- and time orientation-based in the case of goal-directed ST. In detail, Latinjak et al. [3] describe the different subtypes of goal-directed ST, e.g., to control or regulate cognitive reactions, activated states, and motor tasks.

Based on Hardy et al. [1], who postulated the four mechanisms of ST, van Raalte et al. [2] developed a sport-specific model of ST that includes not only the Systems 1 and 2 distinction, but also the reciprocal effects of ST on behavior, and vice versa. In addition, personal and contextual factors are said to influence the two systems [1,2]. Our study model considers both personal and contextual factors and personal variables in correspondence with this model (e.g., gender and trait anxiety), as well as a specific contextual factor (type of sport). System 2 represents ST interventions, and System 1 represents participants’ states (state anxiety and state self-confidence). In addition, we included the factor (coach-rated) performance. 

Several meta-analyses and reviews in the literature have shown that ST may help to improve motor learning and athletic performance. Hatzigeorgiadis, Zourbanos, Galanis, and Theodorakis [5] summarized 32 experimental studies regarding the impact of ST on motor learning/performance. They found a moderate effect of *d* = 0.48 with higher influence on newly learned tasks than on well-known tasks as compared to control groups with no ST. Overall, it can be concluded that ST training clearly facilitates the learning of new motor tasks. In addition, instructional ST seems to be more influential than motivational ST in these learning situations [5]. Tod, Hardy, and Oliver [6] summarized the results of 47 studies with various tasks and measures of performance. Positive ST (e.g., stay cool) proved to be more effective than negative ST (e.g., do not worry), and ST had a greater impact on cognitive (for example, attention) and behavioral measures (for example, motor behavior) than on the other variables (such as emotions). They found no differential effects of instructional and motivational ST on performance, contrary to Hatzigeorgiadis et al. [5]. A number of experimental studies have been published confirming and extending the results on performance variables in various sports since the publication of these meta-analyses. For example, ST has been found to improve the learning of tennis ground strokes [7,8,9], vertical jump performance [10], the shooting accuracy of football players [11], and cycling speed [12]. In addition, ST seems to have a positive impact on physiological variables, as shown with cyclists in a study conducted by Wallace, McKinlay, Coletta, Vlaar, Taber, Wilson, and Cheung [13]. 

In addition to motor learning and performance, several studies focus on the effects of ST on psychological variables. In this regard, after reviewing the literature on ST interventions, Kahrović, Radenković, Mavrić, and Murić [14] (p. 57) concluded that “there are a sufficient number of studies that confirm the positive effects of self-talk”, “improving (…) self-confidence, as well as reducing anxiety in different kinds of sports”. Other studies have found that the implementation of motivational ST can lead to a reduction of anxiety—in particular, cognitive anxiety. In their review, Tod et al. [6] confirmed the anxiety-reducing effects of ST, as did Hatzigeorgiadis, Zourbanos, Mpoumpaki, and Theodorakis [15] in a study involving tennis players. Latinjak et al. [4] studied the effects of different types of ST in anxiety- and anger-eliciting situations, and obtained greater results for positive ST in anxiety-eliciting than in anger-eliciting situations.

The possible impact of ST training on personality variables, such as trait anxiety, volition, and self-efficacy is another important research question. Personality factors have been underrepresented in sport-specific studies on ST [2]. To date, literature on ST training has either not addressed these dispositional variables or has obtained controversial results. This is unfortunate, because it may be expected that, to be successful, ST should be adapted not only to the situation, the task demands, the stated goal, and/or the athlete’s skill level but also to the athlete’s personality/dispositions, such as general self-efficacy.

Finally, the optimum duration of systematic ST interventions is unclear. Whereas, Beneka, Malliou, Gioftsidou, Kofotolis et al. [16] and Weinberg, Miller, and Horn [17] showed positive effects of a one-week ST intervention, Chang, Ho, Lu, Ou, Song, and Gill [18], Hamilton, Scott, and Macdougall [19], and Hatzigeorgiadis, Galanis, Zourbanos, and Theodorakis [20] only found effects after multi-week interventions. Therefore, our study addressed this issue by manipulating the intervention duration of one week versus eight weeks, as it is expected that more global constructs, such as self-efficacy, may need more time to be positively influenced than immediate state variables, such as state anxiety [21,22]. 

Although the number of studies on potential effects of ST has considerably increased, particularly over the last two decades [23], limitations in the research and, consequently, a lack of knowledge still exist. There has been some bias in the samples that have been considered; females and junior elite athletes are underrepresented, whereas males, students, and non-elite participants outnumber other groups [6]. The restrictions in sample characteristics may threaten the external validity of the results, and therefore it seems warranted to extend the groups of participants that are considered, namely to include females and junior elite athletes. Additionally, dependent variables are incomplete and they contribute to further gaps in knowledge. For example, numerous studies have been conducted on motor performance in the learning of new skills [7,8,9,10], whereas the effects of ST on performance in competitions or at least on the performance of elite athletes has only rarely been considered [5], with some notable exceptions [20,24,25]. Increasing ST studies with elite or sub-elite athletes would not only expand our knowledge, but it would also contribute to evidence-based practice in sport psychology with elite athletes [24].

To summarize, our literature analysis reveals that systematic ST may improve motor learning performance and enhance the psychological mindset of participants; however, there is a preponderance of convenient samples and non-elite athletes in these studies. Additionally, we have an incomplete understanding of the impact of ST on the performance of (sub)elite athletes and on certain psychological variables, such as self-efficacy, trait anxiety, and volition. Finally, the duration of ST training remains under scrutiny. Therefore, the novelty of our study consists of the sample composition (male and female competitive athletes), of the mixture of trait (gender, trait anxiety, general self-efficacy, volition) and state variables (state anxiety, performance), and of testing the effects of differential ST training duration.

## 2. Mechanisms of the Effects of ST

A very important question in the ST literature examines why ST influences dependent variables, such as performance and state anxiety. What mechanisms can be differentiated in the ST paradigm? As described in the Introduction, van Raalte et al. [2] developed a sport-specific model of ST that is based on dual-process theories of information processing and on Hardy et al. [1]. The ST model that was postulated by Hardy et al. considers personal and situational antecedents, as well as four mechanisms of ST that can affect performance and psychological aspects (see Figure 1). Personality, belief in ST, and cognitive processing preference comprise the personal antecedents of ST, whereas task difficulty, competitors, coach behavior, and match circumstance represent situational antecedents (for details, see [1]). These antecedents can influence athletes’ ST [1]. First, the behavioral mechanism focuses on techniques and the execution of motor tasks by an athlete. Accordingly, ST that is derived from this mechanism helps to improve motor learning [1,5], enhances the quality of execution [6], and influences automatic behavior [26]. Second, the cognitive mechanism is represented by cognitive processes of information processing, attention control, and decision making, all of which may help athletes to adequately react to environmental stimuli and initiate adequate activity. In particular, the cognitive mechanism may improve the execution of fine (versus gross) motor tasks [5], enhance attentional focus [27], and reduce interfering thoughts [28]. Third, the affective mechanism primarily has an impact on psychological states, such as mood and state anxiety. For example, ST studies have provided evidence for reduced anxiety and anger states through ST [1,26]. Additionally, ST differs, depending on anger-or anxiety-eliciting situations [4]. Finally, the motivational mechanism is reflected in motivational ST, which may influence athletes’ activation and endurance. In competitions, athletes seem to prefer motivational strategies that help them to achieve an optimal level of arousal [29]. However, Hardy et al. [1] point to a lack of research on the effect of ST on motivational variables, such as fatigue or activation, although motivational ST has been an often used and investigated strategy. 

Figure 1 presents an extended version of the model by Hardy et al. [1], to which we added gender and age for personal antecedents. We further extended the model and included evidence-based details to the mechanisms and proven training effects, such as acceptance, routines, systematic knowledge, commitment, familiarity, intrinsic interest, and motivation [20]. In addition to the antecedents and mechanisms of ST, our model further illustrates the proportion of motivational and instructional ST that two conceptual triangles bind (see Figure 1). For example, instructional ST improves motor skills [1] and it can employ attentional cues to focus on the relevant motor task [28]. Therefore, instructional ST can be found with both cognitive and behavioral mechanisms. However, motivational ST can influence emotional reactions [26].

The extended model serves as basis for our intervention study and it is designed to incorporate all four mechanisms of the model, with the dependent variables corresponding to all four mechanisms: The cognitive mechanism is represented in concentration problems and other aspects of cognitive trait anxiety; the motivational mechanism in volition and self-efficacy, the affective mechanism in competitive state anxiety, and the behavioral mechanism in athletes’ skill development. 

We developed an intervention study with one control group, two intervention groups, and three points of measurement based on our literature review and the aforementioned model (Figure 1). The study was conducted with talented male and female adolescent athletes who had obtained either systematic ST training, in addition to their usual physical training (experimental group), or no additional ST training (control group). Additionally, the length of the intervention period differed between the experimental groups. Approximately half received either one week of training or eight weeks of training. 

The primary objective of the study was to investigate the effects of ST training on talented young athletes’ anxiety level, self-efficacy, volitional skills, and performance. A second objective of the study consisted of testing the differential effects of short-term (one week) versus long-term (eight weeks) interventions. The main hypothesis predicted lower competitive state anxiety, higher volitional skills (e.g., self-optimization), increased self-efficacy, and better performance after ST training than after no ST training. The second hypothesis predicted higher effects in these dependent variables after long-term ST training when compared to short-term training. No differences were expected for competitive trait anxiety, as it is a trait variable that ST training should not directly affect. Finally, no differential effects of the intervention were expected for male and female participants. Both variables, trait anxiety and gender, were chosen to contribute to a better understanding of the possible impacts of personal factors on the ST–behavior relationship.

## 3. Materials and Methods

### 3.1. Participants

The sample consisted of 117 participants that were aged 13 to 23 years (*M* = 16.0, *SD* = 1.8; 55 females, 62 males), recruited from a variety of individual sports (*n* = 72), such as canoe racing, gymnastics, rhythmic gymnastics, swimming, and wrestling, and from team sports (*n* = 45), such as ice hockey, handball, and volleyball (Table 1). The participants were students at sports schools or athletes in sports clubs in three different cities. The criteria for including participants were (at least) 12 years of age and participation in a competitive sport activity. Both criteria were chosen to capture the thus far underrepresented sample of junior sub-elite athletes. 

On average, the athletes were active in their respective sports for approximately 8.7 years (*SD* = 2.8) and had three to 18 training sessions per week (*M* = 8.7, *SD* = 3.3), with an average of 16.2 hrs per week (*SD* = 6.2). At the time of the study, 109 participants regularly took part in competitions; of these, *n* = 33 team sport athletes and *n* = 56 individual sport athletes were in the national C/NK1-, DC/NK2-or D-squads (According to the athletes’ age and level of performance they are assigned to five squads: A/OK, B/PK, C/NK1, DC/NK2 and D. The C/NK1, DC/NK2 and D-squads are for junior aged athletes, whereas the A/OK and B/PK-squads are for (mature) elite athletes) or in the highest to third-highest competition levels. Thus, 89 athletes can be referred to as junior elite athletes, according to the categorization of Rees, Hardy, Güllich, Abernethy, Côté, Woodman, et al. [30].

In total, 13 participants withdrew for various reasons after the first (Time 1) measurement and prior to the onset of the study. In the experimental groups, 36 athletes in the short-term group and 38 athletes in the long-term group completed the ST intervention. In the control group, 30 participants took part in the study and completed the questionnaire measures. As a result, the experimental and control groups consisted of 104 athletes. The number of participants in the statistical analyses decreased to 95 participants at Time 2, and 80 at Time 3 due to missing data on some dependent variables (Figure 2). 

### 3.2. Design and Procedure

Permission to conduct the study was obtained by the university’s ethics committee (file number: 325/16-ek). Different sport associations and one elite sport school were contacted and informed about the study. In the elite school, a project kick-off meeting with teachers and coaches was organized to present the study. Athletes were contacted in the respective training groups and asked if they would volunteer to participate after the coaches granted consent for the study. The participants were informed about the study objectives and were assured of anonymity and confidentiality in the recording, analysis, and presentation of data. The parents of all underage athletes were also informed regarding the study objectives and procedures and were asked for their permission. After giving informed consent and with the permission of their parents, the participants completed the questionnaire measures of the dependent variables (Time 1). The research questionnaires were distributed at regularly scheduled training sessions and they were completed under the supervision of one of the authors, and in the absence of the coaches. Afterwards, the participants were blindly randomized to either the control group or to one of the experimental groups. Participants in the short-term experimental group received one week of ST intervention, including three 60-min sessions, whereas the long-term experimental group received eight weeks of ST training, with three 20-min sessions per week. The coaches were not informed about the group assignment to prevent influences on athletes’ behavior.

After finalizing the intervention program, the participants of the intervention groups and the control group again completed the dependent measures (Time 2). Half of the control group were tested after one week, and the other half were tested after eight weeks. Finally, to test whether the treatments had any lasting effects, we conducted a third data assessment five to six weeks later, depending on the athletes’ time schedule (Time 3). The control group participants were tested parallel to the experimental groups.

### 3.3. Intervention

#### 3.3.1. Short-Term and Long-Term Intervention Groups

The ST intervention in this study aimed to help athletes individually tailor their ST, such that it should positively influence levels of anxiety, self-efficacy, volitional skills, and performance. The ST that was developed in this study is understood as goal-directed and self-determined, aiming to focus, control, or regulate cognitive reactions, activated states, or relevant motor tasks [3,31].

The contents of the intervention program were based on the modified ST model that was proposed by Hardy et al. [1] (see Figure 1). In detail, athletes became familiar with the four mechanisms of ST, that is, behavioral, cognitive, affective, and motivational mechanisms. In the first session, the participants received information regarding the possible objectives and effects of systematic ST and discussed their individual understanding of ST. In preparation for the different types of ST, the participants also recorded ST words and phrases that they had used in the past. After every introduction to a mechanism, the athletes were asked to develop the possible ST for the respective mechanism to develop their own ST. After the athletes developed their individualized ST, focusing on one mechanism, they were asked to practice it during the training sessions. The participants were asked for feedback at the end of the ST intervention and to record their individually developed ST (see the paragraph about recording and evaluating athletes’ ST). Table 2 presents an overview of the intervention’s contents.

The program’s duration differed between short- and long-term intervention groups. The short-term intervention group received three 60-min ST training sessions within one week, whereas the long-term intervention group received three 20-min sessions per week for eight weeks.

In the first session, the short-term intervention group learned the four mechanisms of ST, focusing on different questions for each mechanism: behavioral: ‘How do I optimize movement?’; cognitive: ‘How do I improve my attention?’; affective: ‘How do I regulate my emotions?’; motivational: ‘How do I modify my motivation?’ (see Table 2). At the end of the session, the participants reflected on the possible keywords that an individual ST could have for every mechanism. In the second session, the athletes were introduced to breath control strategies to determine the effects of ST on thoughts, emotions, movement, and attention during competitions. Furthermore, the athletes reviewed the four mechanisms of ST and tried to develop their own individual ST for the mechanisms. In the last session, the athletes had to choose one ST that they developed in the last two sessions and had to work on it.

Athletes in the long-term intervention group received a deeper introduction to the topic of ST and had to record their previously used ST in the first week. In the second week, the athletes learned breath control strategies (see short-term intervention group). In the third week, the participants focused on the behavioral mechanism and developed ST to influence activation or to improve motor-learning skills, to improve on potential movement deficits. The fourth week focused on the cognitive mechanism, where the athletes recorded their individual focus during training and competition. They developed ST to remain focused on performance details and to avoid distractions. In week five, the athletes learned the affective mechanism of ST and had to reflect on their positive and negative emotions during training and competitions and consider what type of ST helps to regulate these emotions. In week six, the long-term intervention program focused on the motivational mechanism, and the athletes developed strategies to influence their motivation while using ST. Finally, in weeks seven and eight, the athletes chose one ST that they had developed in prior sessions to employ for their training sessions and competitions.

#### 3.3.2. Control Group

The control group had no ST training during the study. Whereas the participants in the short- and long-term interventions had the ST intervention prior to or after their physical training sessions or during a scheduled time, athletes in the control group had no additional appointments. They received an ST intervention after the end of the study for ethical reasons and for reasons of equal treatment.

#### 3.3.3. Recording and Evaluating Athletes’ Self-Talk

As mentioned above, the athletes recorded ST words and phrases that they had used in the past. The athletes were again asked to record their ST words at the end of the intervention to understand the potential development of these ST words and phrases and to assess whether their ST was goal-directed. This type of ST evaluation was conducted in interview or group discussion format, and the results were collected in written form.

### 3.4. Measures

Competitive state anxiety and competitive trait anxiety were assessed while using the Competitive Anxiety Inventory State (WAI-S) and the Competitive Anxiety Inventory Trait scales (WAI-T; adapted by Brand et al. [32]). The WAI-S is the German version of the Competitive State Anxiety Inventory-2 (CAI-S-2) that was developed by Martens, Vealey, and Burton [33] and makes a distinction between somatic and cognitive dimensions of competitive state anxiety. The WAI-S consists of three subscales (somatic state anxiety, cognitive state anxiety, and state self-confidence) with four items each. Participants are asked to indicate “how you feel right now” for each item on a four-point Likert scale ranging from 1 (*not at all*) to 4 (*very much so*). Examples of the somatic state anxiety items include “I feel tense in my stomach” and for cognitive state anxiety “I’m concerned about performing poorly”. These items differ from the positive component, state self-confidence, for example, “I’m confident of coming through under pressure”. The WAI-S has been used in other studies [34,35] and it has been shown to be a valid instrument for athletes with sufficient internal consistency, with a Cronbach’s α ranging from 0.79 to 0.82 [36]. 

In contrast to the WAI-S, the trait version (WAI-T) assesses an athlete’s disposition to perceive competitive situations as threatening. The questionnaire is a German adaptation of the Sport Anxiety Scale (SAS), as developed by Smith, Smoll, and Schutz ([37]; adapted and validated by Brand et al. [32]). The participants are asked to indicate how they typically feel “prior to or during competition” [32] (p. 266, footnote 1). Similar to the WAI-S, the WAI-T includes three subscales, one for somatic trait anxiety (e.g., “My heart pounds before a competition”) and two for cognitive trait anxiety. These two subscales measure worry (e.g., “I am concerned about choking under pressure”) and concentration disruption, which represents the athlete’s disposition to being agitated by confusing thoughts in competitive situations (for example, “While performing, I often do not pay attention to what’s going on”). The participants respond to three subscales, according to a four-point Likert scale ranging from 1 (*not at all*) to 4 (*very much so*). The WAI-T has been used with athletes in other studies [35], and its internal consistency is regarded as sufficient (a Cronbach’s α ranging from 0.77 to 0.82; [32]). 

The summed scores of the WAI-S and the WAI-T items represent the level of intensity that the athlete is feeling with regard to somatic and cognitive state anxiety and to state self-confidence, on the one hand, and somatic trait anxiety, worry, and concentration disruption, on the other hand. Competitive trait anxiety can be regarded as a disposition to answer with a heightened level of anxiety in competitive situations in sport and it was included as a personal characteristic that might influence state anxiety reactions. State anxiety as main dependent variable was included, because we expected a significant impact of the ST intervention on state anxiety. 

Volition was measured while using the Volitional Components Questionnaire Sport (VCQ-Sport; [38]). The VCQ-Sport is based on the Volitional Components Inventory by Kuhl and Fuhrmann [39]. The questionnaire consists of 60 items in four subscales (self-optimization, self-impediment, energy deficit, and loss of focus) that measure volitional skills and deficits that are related to training and competitions. Self-optimization includes strategies that help to realize intentions and achieve goals (e.g., “I have an optimistic attitude towards most aspects related to sports”). Self-impediment refers to negative thoughts and worries regarding possible failure and can be represented by the item “When I am upset/have negative feelings while playing a sport, it is difficult for me to adjust my attitude”. Energy deficit corresponds to a lack of energy and activation, which may lead, for example, to the avoidance of training hours or to a-motivation (for example, “During training, I often lack sufficient energy”). Loss of focus includes a lack of attention to relevant stimuli in training and competition, such that the athlete may be concerned with distracting stimuli or may avoid competitions. This scale includes items, such as “I sometimes have to motivate myself to compete”. All of the items were answered on four-point Likert scales, ranging from 0 (*not at all true*) to 3 (*very true*). Scores for each subscale were calculated by adding the item values. Higher scores in self-optimization and lower scores in the other three scales typically indicate positive volitional skills in high-performance athletes [38]. The VCQ-Sport shows acceptable internal consistency, with a Cronbach’s α ranging from 0.76 to 0.92 [38], and it has been used in other studies [40].

Self-efficacy was assessed while using the German version of the General Self-Efficacy Scale (GSE; [41]; validated by Hinz, Schumacher, Albani, Schmid, and Brähler [42]). We refrained from using sport-specific self-efficacy instruments to make the data comparable for all athletes, independent of sport type, due to the variety of sports included in our study. The participants were asked to answer items, such as “Thanks to my resourcefulness, I can handle unforeseen situations” or “I am certain that I can accomplish my goals” on a four-point Likert scale, ranging from 1 (*not at all true*) to 4 (*exactly true*). The GSE score reflects the strength of an athlete’s self-efficacy belief and it is constructed by adding all of the responses to a total score that may range between 10 and 40, with higher GSE scores indicating higher self-efficacy. Internal consistency with a Cronbach’s α ranges from 0.76 to 0.90 in various studies [42,43]. The GSE scale was included as a valid and internationally known instrument that has been used in a variety of settings, including exercise and sport [43,44]. In addition, the scale has been tested with a representative German sample, and our results can thus be compared to population norms [42]. 

Athletes’ current performance was evaluated according to the procedure that was used by Alfermann, Lee, and Würth [45]. Each athlete’s coach was asked to rate the athlete’s skills on eight rating scales. As a result, each athlete was rated on the same eight rating scales by his or her coach with regard to endurance, fitness, strength, coordination skills, mental skills, and tactical competences. Items, such as “Currently, the athlete can handle endurance challenges easily”; or “Currently, the athlete is able to show the necessary coordination skills” were rated on five-point Likert scales from 1 (*very low*) to 5 (*very high*). The composite performance score was calculated by adding the scores of all eight items, thus varying from 8 to 40. For the current study, we conducted a reliability analysis for Time 1 to estimate the internal consistency of the eight items’ performance scale. A Cronbach’s α = 0.78 is regarded as an acceptable value.

### 3.5. Data Analysis Plan

Statistical analyses were conducted while using the Statistical Package for the Social Sciences (SPSS) version 24. Univariate analyses of variance and Chi-square test were conducted to examine the possible sociodemographic and sports-related differences between the three groups (short-term intervention, long-term intervention, control group). Multivariate and univariate analyses of variance for all of the dependent variables were conducted to investigate the differences between male and female athletes in the dependent variables at Time 1. The *p* value was restricted to *p* < 0.01 to avoid alpha error inflation. Partial *η*^2^ values were included, in addition to the usual *p* values. The partial *η*^2^ is the proportion of the total variability that is attributable to a factor, and it can thus be regarded as an estimate of the effect size. We relied on Cohen’s suggestions [46] that *η*^2^ = 0.01 represents a small effect size, 0.06 a medium effect size, and 0.14 a large effect size when interpreting the *η*^2^ values. Multivariate analyses were conducted for each construct measured while using a questionnaire with more than one subscale (WAI-S, WAI-T, and VCQ-Sport). Bonferroni post-hoc tests were calculated to explore the structure of the effect in the case of an overall significant effect on the dependent variables.

## 4. Results

Univariate analyses of variance revealed no significant group differences regarding age (*p* = 0.163), years in sport (*p* = 0.751), training sessions per week (*p* = 0.482), and training hours (*p* = 0.760). Furthermore, the Chi-square test revealed no significant differences regarding the competition participation (*p* = 0.452), competition level (*p* = 0.610), or squad category (*p* = 0.612).

### 4.1. Statistical Analyses for Time 1

First, the differences between male and female athletes in the dependent variables at Time 1 were addressed. A multivariate main effect of gender was found for WAI-S, m*F*(3, 113) = 4.15, *p* = 0.008, and *ƞ*^2^ = 0.10. The following univariate analyses proved to be insignificant for somatic state anxiety, *F*(1, 115) = 0.42, *p* = 0.52, and *ƞ*^2^ = 0.004, but significant for cognitive state anxiety, *F*(1, 115) = 8.20, *p* = 0.005, and *ƞ*^2^ = 0.08, as well as state self-confidence, *F*(1, 115) = 9.50, *p* = 0.003, and *ƞ*^2^ = 0.09, with higher mean scores for the females in cognitive state anxiety (*M*_F_ = 9.19, *M*_M_ = 7.68) and lower scores in state self-confidence (*M*_F_ = 9.15, *M*_M_ = 10.61). Both of the differences can be regarded as moderate. No gender differences emerged in WAI-T and VCQ-Sport, m*F*(3, 113) = 2.94, *p* = 0.04, and *ƞ*^2^ = 0.07; and m*F*(4, 112) = 2.65, *p* = 0.04, and *ƞ*^2^ = 0.09, respectively. For general self-efficacy, males scored higher than females (*M*_M_ = 27.87, *M*_F_ = 25.76), *F*(1, 115) = 7.25, *p* = 0.008, and *ƞ*^2^ = 0.06, which represents a moderate difference. Finally, performance was not differently rated by the coaches for male and female athletes. Analyses of the intervention effects on self-efficacy and state anxiety considered the possible differences between male and female athletes. The data for males and females were combined, as there were no interaction effects of Group × Gender on those variables. 

In a second step, we looked for the possible differences between the 104 participants who remained in the study and the 13 participants who dropped out of the study after Time 1 (before the assignment to the intervention or control groups). This was done to check whether the later dropouts might have an impact on the study’s data. The dropouts and the two intervention groups scored higher on self-impediment than the control group. This was the only significant result (*F*(3, 113) = 3.72, *p* = 0.01, *ƞ*^2^ = 0.09). This means that the dropouts were in general not different from those who remained in the study. Table 3 presents the descriptive statistics for all variables.

Differences between experimental and control groups at Time 1 were addressed in the same manner as done for gender differences with multivariate and univariate analyses of variance. There were no significant differences in the dependent variables between the control group and the short-term experimental group or between the control group and the long-term experimental group, with one exception (self-impediment). Therefore, the assumption seems justified that the randomization of the participants into the three groups was successfully completed and the groups did not differ in the dependent variables before the start of the experiment. 

### 4.2. Statistical Analyses of Intervention Effects: Results of 3 (Time) × 3 (Group) Analyses of Variance for Pre-, Post- and Follow Up-Measures

The effects of the intervention program on the dependent variables were first addressed with multivariate two-way analyses of variance with one within (time) and one between (group) subject factor. The time variable included the three measures, before the intervention, after the intervention, and a follow-up. The group variable included the three categories (two intervention groups and one control). Descriptive statistics and results of the various analyses of variance can be found in Table 3. State anxiety, trait anxiety, and volitional skills were first tested with separate multivariate, and then univariate, 3 × 3 analyses of variance. Self-efficacy and performance were each analysed with separate univariate analyses of variance. There were no significant main group effects in all of the analyses. The multivariate analysis for trait anxiety revealed no interaction effect, but only a significant main effect of time (Wilks Lambda = 5.23; *df* = 6, 304; *p* < 0.001; *ƞ*^2^ = 0.09), which could be traced back to all three subscales. This means that trait anxiety improved on all three subscales during the study duration, regardless of group assignment. 

We found the expected significant interaction effects of Time × Group, apart from significant main effects of time for self-efficacy and performance ratings. With regards to self-efficacy, pairwise analyses for each of the three groups showed a significant increase for the long-term intervention (LTI) group from Time 1 to Time 2 (*p* < 0.001), and from Time 1 to 3 (*p* < 0.001), but no significant increase between Times 2 and 3. The short-term intervention (STI) group improved in self-efficacy between Times 1 and 2 (*p* = 0.04); however, the effect did not remain stable until Time 3. There were no significant differences between the three points of measurement for the control group (CG). Similarly, with regard to the performance ratings, we found significant differences in performance for the LTI group between Times 1 and 2 (*p* < 0.001), as well as between Times 1 and 3 (*p* = 0.01), but not between 2 and 3. There were no significant differences in the performance of STI and CG between the three points of measurement. This means that the LTI group profited from the intervention, whereas the other two groups did not improve in performance. 

Finally, the expected multivariate interaction effects were found for state anxiety and volitional skills, with no additional main effects of time (Table 3). Univariate analyses of variance for state anxiety revealed interaction effects on somatic state anxiety and state self-confidence, whereas cognitive anxiety proved to be nonsignificant. Pairwise comparisons revealed that both of the intervention groups decreased in somatic state anxiety after the intervention, and even beyond, whereas the control group showed a slight, marginally significant increase from Time 1 to 2 (*p* = 0.07). The STI group decrease from Time 1 to 2 was marginally significant (*p* = 0.07) and significant from Time 1 to 3 (*p* = 0.04), and the LTI group from Time 1 to 2 was only marginally significant (*p* = 0.07). The following pattern of results emerged with regard to state self-confidence: for the STI group, there were no significant changes in self-confidence over time, whereas, for the LTI group, self-confidence increased from Time 1 to 2 (*p* = 0.03) and again from Time 2 to 3 (*p* = 0.005). For the control group, state self-confidence unexpectedly increased from Time 1 to 2, marginally significant (*p* = 0.08), but then dropped significantly until Time 3 (*p* = 0.006).

The analysis of volitional skills (VCQ-Sport) revealed a significant multivariate interaction effect Time × Group, which only depended on the self-optimization scale. No other volitional skill variables showed significant effects, which means that the participants’ volitional skills did not change over time or depending on the intervention. Pairwise comparisons of the means show the expected increase in self-optimization for the LTI group (Time 1 to 2: *p* = 0.004; Time 2 to 3: *p* = 0.003), but no significant changes for the STI group over time and a decrease in self-optimization on part of the control group (Time 1 to 3: *p* = 0.004; Time 2 to 3: *p* = 0.04). This means that the ST intervention improved the ability of the athletes in the long-term intervention group to set and follow their goals, even beyond the end of the intervention, whereas the short-term intervention group stagnated and the control group participants significantly lost self-optimization skills during the whole study.

## 5. Discussion

This study tested the possible effects of junior sub-elite athletes’ ST on psychological variables and performance ratings. To this end, the athletes who volunteered for our study were randomly assigned to one of three conditions: (1) a control group without ST training; (2) a short-term ST intervention group in which the athletes learned positive ST in three 60-min sessions in one week; and, (3) a long ST intervention group in which the athletes received three 20-min sessions of ST training per week for eight weeks. Our two main hypotheses first focused on the effects of ST training versus no training on athletes’ state and trait competitive anxiety, volitional skills, general self-efficacy, and performance, and, second, on the effects of long-term training as compared to short-term training. As expected, we found no gender differences in the reaction to the ST training and no differences between dropouts and non-dropouts in our study. As per the first hypothesis, we compared the scores before the intervention, after the intervention, and again in a follow up, and found the expected significant interaction effects of Time × Group on somatic state anxiety, state self-confidence, self-optimization (a volitional skill), self-efficacy, and performance as rated by the coaches. 

When summarizing these results, we can conclude that the ST intervention influenced the participants more than the control group, but not on all the dependent variables. As expected, the intervention did not affect trait anxiety. However, unexpectedly, trait anxiety improved over time for all participants, regardless of group assignment. This may be explained as an indirect effect of our experiment on all participants, or it can be explained as a learning effect over time due to an increasing adjustment to competitions in all participants or due to an adjustment in completing our trait anxiety measurement instruments. As our participants engaged in their usual training sessions and competitions during our study, an additional explanation of the main effect of time co33uld be that all athletes simply matured psychologically. This explanation seems even more likely when considering the age range of most of our participants, specifically, adolescence. Alternatively, the main effects of time may be due to the Hawthorne effect on the part of our control group participants. Knowing that they belonged to a group of ‘selected’ athletes (who, in addition, had been promised the same treatment as the experimental participants after the end of the study) may have influenced their reactions in the direction of the experimental participants. Additionally, we cannot exclude the possibility that the athletes in the experimental and control groups had not only been in contact with each other, but had also shared their knowledge regarding our study in general, and the ST intervention in particular. This means that our control participants could also have practiced ST strategies during the project by learning from the experimental participants, which, as a result, could have biased their answers on the trait anxiety scales. However, it seems unlikely that this bias would only happen for one variable. Therefore, we tend to reject this assumption.

As stated above, the expected interaction effects were found for five variables (state somatic anxiety, state self-confidence, self-efficacy, self-optimization, and rated performance), whereas the intervention did not affect cognitive state anxiety and three volitional skills. Before turning to the significant results on the five variables, we would like to speculate on the reasons why cognitive state anxiety and three of the four volitional skills remained uninfluenced by ST training. First, the trait-like variables may be less specifically changeable. This may help to explain the results with no effects for the volitional skills of self-impediment, lack of energy, and loss of focus. However, self-optimization, which can also be regarded as a trait-like variable, significantly and considerably improved, but only for the LTI group. The reason for the different effects on the four scales may be found in the item wording. The self-optimization items are all positively worded (with words such as take initiative; intrinsic motivation; positive thinking), whereas the items of the other three VCQ-scales are negatively worded—with words, such as lose energy; unmotivated; dislike exhausting work. This could have contributed to an answering bias of social desirability. As most of the participants were sub-elite adolescent athletes, they would tend to more easily agree to self-optimization items; however, they would instead only reluctantly admit a loss of motivation and energy in training and competition. As a consequence, the scores tended to be low for most of the participants on these three negatively phrased scales, irrespective of group assignment and time of measurement. The participants’ mean scores on self-impediment, lack of energy, and loss of focus were below average, as compared to norms that were reported in the scoring book of the test authors [38]. In addition, as the ST sentences learned during the intervention phase were also positive—such as the items of the self-optimization scale—this active and intrinsically motivated behavior may have been more easily and ‘logically’ a result of ST learning in the eyes of the participants. This means that the non-significant results for three of the four VCQ scales could possibly be traced back to the scale construction, which may have prevented our participants from more strongly agreeing to the items. 

In addition to the non-effect results on the three VCQ-scales, we also found that cognitive state anxiety did not change over the course of our study in any of the three groups. This came as a surprise to us, and we will discuss this result in combination with the two other scales of the WAI-S questionnaire: somatic state anxiety and state self-confidence. 

Cognitive state anxiety and state self-confidence have been identified in earlier studies as being influenced through ST [6,14,15,47]. Thus, these two variables seem well-suited to indicate a positive development in athletes after ST intervention. Our study corroborates the findings on self-confidence and contributes to the literature by expanding the external validity of prior research. Our participants were competitive male and female athletes, most of them members of squads, and thus our results show the generalizability of the effect of ST training on state self-confidence to a wider range of participants. Additionally, the effects on self-confidence could only be found for the LTI participants whose scores increased after the ST training and even beyond, which means that the intervention had a sustainable effect for the LTI group, whereas the STI participants’ self-confidence was not affected. Obviously, the short ST intervention was not influential enough to affect the athletes’ self-confidence.

Although the results on state self-confidence are somewhat congruent with the literature, the results on state anxiety are contradictory. On the one hand, we find an interaction effect of Time × Group on somatic state anxiety, which at least does not contradict the former results; however, on the other hand, we find no effects regarding cognitive state anxiety. This is striking, as the latter variable has been confirmed in earlier studies of ST, for example, in the review of Tod et al. [6] (p. 677): “All studies examining cognitive anxiety reported a beneficial effect of ST, whereas 75% of studies examining somatic anxiety showed no effect”, but 25% did! At first glance, it seems that our ST intervention, which obviously influenced somatic but not cognitive state anxiety, contradicts the literature. However, at second glance, our study may help to clarify the influenceability of somatic state anxiety. Tod et al. [6] (p. 677) concluded that “The existing evidence regarding somatic anxiety demonstrates no clear effect”. In our case, both intervention groups scored significantly lower on somatic state anxiety after ST training than before, the short-term intervention group decreased even further at Time 3, and the long-term group remained equal to Time 2, which thus supported the sustainability of the effect. As somatic state anxiety represents physiological signs of anxiety, it is conceivable that our ST intervention had a positive influence on physiological aspects. Barwood et al. [12] found that motivational ST enabled a higher power output (VO_2_, W). We may have had similar effects without physiological proof. Regarding cognitive anxiety, unfortunately we cannot support existing evidence of influenceability through ST, and thus we can only recommend that further studies be run to test the expected effect of ST training on cognitive state anxiety. 

ST training also influenced coach-rated performance, which shows that it cannot only have an impact on certain motor behaviors, but also on more general motor skills, as rated by the coaches. However, the results show this effect for the LTI group only and in addition during the whole duration of the experiment, whereas the STI group’s performance did not profit from ST. Again, as with self-confidence and self-optimization, the long-term intervention proved to be superior to the short-term intervention with regard to its impact on the participants. Our measure of performance reflects athletes’ athletic and mental skills, as seen by the coaches, and can therefore be regarded as a more general, composite skill. Obviously, a longer duration of ST training is needed to change that skill. The short-term intervention could not fulfil the necessary demands for a performance change. Self-efficacy, for which influenceability through ST is controversial in the literature, reacted positively to ST training in both intervention groups, but more in a long-lasting way in the long-term group. Although both intervention groups improved at Time 2, only the LTI group remained at the same level at Time 3; and, the STI group deteriorated. Finally, self-optimization, which is an important volitional skill of athletes that helps them to attain their goals and stay motivated, increased after ST training for the LTI group, whereas the STI group did not gain in self-optimization during the course of the experiment. In contrast, the control group even lost self-optimization skills at Time 2, and this was further decreased at Time 3. This means that ST training helps to keep athletes motivated and eager to train and compete (STI group), or even increase their motivation (LTI group), whereas the absence of ST training results in the loss of motivation over time. Volitional skills have yet to be considered in the ST literature, and thus our results may offer new insights and suggestions for further research. 

Our second hypothesis, which predicted greater effects for the long-term than for the short-term intervention, was partially supported. As described above, the long-term intervention was more successful (in terms of impact and sustainability) than the short-term intervention on four of the five significant effect variables, specifically, state self-confidence, self-efficacy, self-optimization, and coach-rated performance. The only slight exception is somatic state anxiety, for which the STI group continuously decreased from Time 1 to 3, and the LTI group from Time 1 to 2 only, but remained stable until Time 3. This means that, overall, short-term ST training was less effective than long-term training, but there are nevertheless significant desirable effects on self-efficacy and state anxiety. Additionally, the STI group (and the LTI group alike) developed better than the control group, which did not show improvements on our dependent measures.

Our third hypothesis predicted no gender differences in the dependent variables and in the intervention effects. In the dependent variables at Time 1, we found small-to-moderate gender differences in three variables, specifically, cognitive state anxiety, state self-confidence, and self-efficacy. These differences seem to make no contribution to our main hypothesis, because, as expected, we did not find any differential intervention effects in both genders. Nevertheless, future studies should control for possible gender differences of ST intervention effects to expand the external validity of studies.

## 6. Limitations

Different sport associations and elite sport schools were contacted and informed about the study to gain access to sub-elite athletes in our study. Teachers and coaches then informed their athletes and provided contact details. Thus, only coaches who were interested in sport psychology in general and in ST in particular were likely to share this information.

Another limitation of our study is related to the impact of the coaches. Although the coaches had no information about the group assignment, they were aware of the athletes’ participation. It cannot be ruled out that this knowledge may have influenced the coaches’ interactions with their athletes and, in turn, their performance ratings. In future studies, we suggest that a third party be responsible for evaluating the athletes. We also suggest the use of hormonal markers (i.e., testosterone, cortisol, etc.) to evaluate the athletes ‘performance. However, in an applied setting, coaches should be informed regarding the group assignment to support the sport psychologists’ work. In this context, future research could establish additional measurements for all groups, e.g., in weeks 0, 2, 8, and 14. Roberts et al. also suggest an assessment of short-term intervals to track personality trait changes [22].

A potential spill over effect from the intervention group to the control group was minimized by asking the participants of the intervention group not to talk about the study in detail. However, it cannot be ruled out that the athletes in the control group became aware of the content of the intervention.

Another limitation could be the heterogeneity of the sports that were included in the study, where the participants competed in eight different types of sports. It is likely that the different types of sports demand different psychological skills, and thus influence different psychological aspects. Future studies should focus on a smaller number of sports types to reduce the heterogeneity of the sample.

## 7. Applications

The applied aim of this study was to help junior sub-elite athletes to develop an individualized ST. The athletes were asked to report previously used ST words and phrases before the ST intervention and again at the end of the ST intervention to understand the potential development of ST and to assess whether the ST is goal-directed. These data were assessed through interviews and group discussions. Most of the athletes had emotionally driven ST (i.e., system 1) at the beginning of the intervention. By focusing on one mechanism following the extended model that is presented in Figure 1, the athletes were able to develop more directed, consciously driven ST (system 2). For the affective mechanism, for example, one athlete changed ST from ‘I will not lose’ to ‘I have trained a lot and I am well prepared’. Another athlete chose the behavioral mechanism and the ST from ‘What is waiting for me here?’ to ‘Remember the technique: long step, short step, jump’. Summarizing our study from a practical point of view, the ST intervention helped athletes to customize their ST according to their individual needs. The athletes of the intervention groups became more conscious of their emotionally driven ST, and thereby increased their goal-directed, consciously driven ST. Following the ST model that was postulated by van Raalte et al. [2], it can be postulated that the athletes transferred their ST from System 1 to System 2. 

The implementation of this study was limited by the athletes’ well-planned schedule of study hours and daily training sessions. That is, adolescent athletes in our study engaged in both sports and education according to strict schedules. Moreover, scheduling appointments for the short-term intervention group (three 60-min sessions in one week) as well as the long-term intervention group (three 20-min sessions per week for eight weeks) were problematic for the athletes. This challenge was exacerbated for the long-term intervention group, as their ST intervention was directly held before or after their physical training session. The rapid scheduling changes between the ST intervention and physical training, and vice versa, were criticized by the athletes, whereas members of the STI group were satisfied with the schedule. In this case, greater satisfaction may have led to better acceptance of the short-term intervention. Additionally, the STI group developed better than the control group, who did not show improvements in the dependent measures. Thus, in terms of time constraints, the short-term intervention can be recommended, particularly when compared to no ST intervention. However, regarding the overall findings, we strongly suggest regular long-term sport psychological support for young athletes to foster cooperation and trust, and ultimately achieve better results. 

Together, these findings suggest that sport psychological training sessions or intervention programs should be held temporally independent from physical training sessions for maximum effectiveness. Although sessions should be conducted in proximity to physical training, they should directly take place before or after the athletes’ physical training sessions. Furthermore, we suggest a limit of no more than two appointments per week at 60 min per session. In this way, the athletes can participate in their sport psychological sessions fully recovered and may be more focused on the presented content.

## Figures and Tables

**Figure 1 sports-07-00148-f001:**
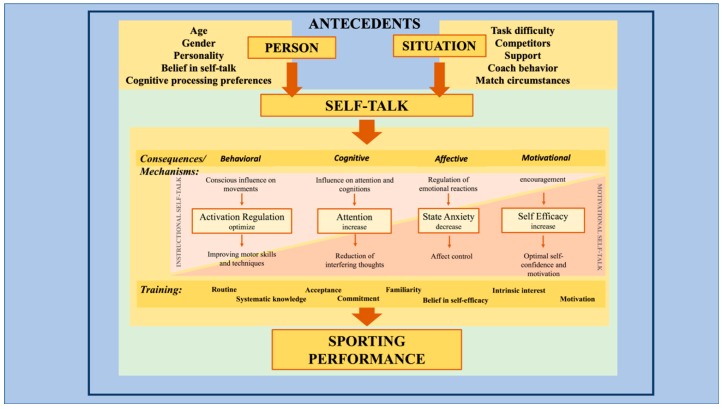
Behavioral, cognitive, affective, and motivational mechanisms of self-talk (modified model, based on Hardy et al., 2009).

**Figure 2 sports-07-00148-f002:**
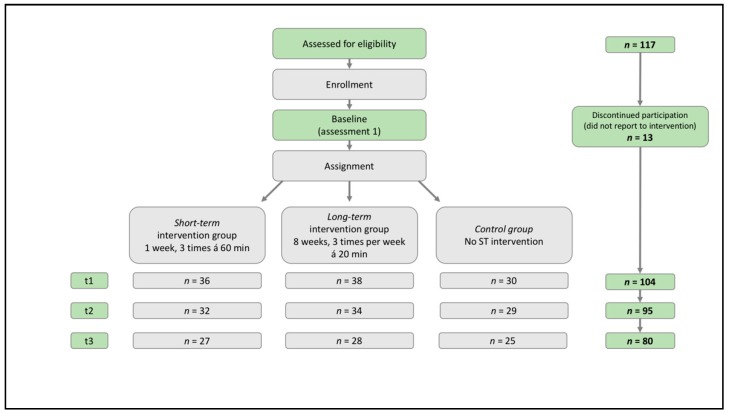
Intervention study design. Participant assignment to the short-term and long-term intervention groups, and the control group, flowchart for Time 1 (t1) to Time 3 (t3).

**Table 1 sports-07-00148-t001:** Sample description for total sample, short-term intervention (STI), long-term intervention (LTI), and control group (CG), dropout *n* = 13 (11.1%).

	Total Sample (*n* = 117)	STI (*n* = 36)	LTI (*n* = 38)	CG (*n* = 30)
**Sociodemographic variables**	*M* (*SD*)			
Age (years)	16.0 (1.8)	15.7 (1.9)	15.7 (1.5)	16.3 (1.6)
Years in sport	8.7 (2.8)	8.4 (3.1)	8.8 (2.7)	8.9 (2.5)
Training sessions per week	8.9 (3.3)	9.1 (3.7)	9.1 (3.2)	8.2 (3.4)
Hours of training per week	16.2 (6.2)	15.8 (5.9)	16.8 (6.7)	16.2 (5.6)
**Gender**	*n* (%)			
Male	62 (53)	20 (55.6)	23 (60.5)	17 (56.7)
Female	55 (47)	16 (44.4)	15 (39.5)	13 (43.3)
**Type of sport**				
Individual sports				
Canoe racing	11 (9.4)	5 (13.9)	4 (10.5)	2 (6.7)
Gymnastics	2 (1.7)	1 (2.8)	1 (2.6)	-
Rhythmic gymnastics	6 (5.1)	1 (2.8)	2 (5.3)	2 (6.7)
Swimming	20 (17.2)	6 (16.7)	6 (15.8)	5 (16.7)
Wrestling	16 (13.7)	5 (13.9)	5 (13.2)	6 (20.0)
Judo	17 (14.5)	7 (19.4)	5 (13.2)	5 (16.7)
Team sports				
Ice hockey	18 (15.4)	7 (19.4)	8 (21.1)	3 (10.0)
Handball	15 (12.8)	1 (2.8)	3 (7.9)	3 (10.0)
Volleyball	11 (9.4)	3 (8.3)	4 (10.5)	4 (13.3)
**Competition participation**	*n* (%)			
Seldom	2 (1.7)	-	1 (2.6)	1 (3.3)
Sometimes	5 (4.3)	1 (2.8)	1 (2.6)	3 (10.0)
Regularly	109 (93.2)	35 (97.2)	36 (94.7)	26 (86.7)
Missing	1 (0.9)	-	-	-
**Competition level (division)**	*n* (%)			
Highest level	30 (25.6)	6 (16.7)	12 (31.6)	9 (30.0)
Second highest level	21 (17.9)	7 (19.4)	7 (18.4)	6 (20.0)
Third highest level	38 (32.5)	10 (27.8)	9 (23.7)	10 (33.3)
Fourth highest level	12 (10.3)	4 (11.1)	5 (13.2)	3 (10.0)
Other level	12 (10.3)	7 (19.4)	4 (19.5)	1 (3.3)
Missing	4 (3.4)	2 (5.6)	1 (2.6)	1 (3.3)
**Squad**	*n* (%)			
A/OK	1 (0.9)	1 (2.8)	-	-
C/NK1	6 (5.1)	1 (2.8)	1 (2.6)	2 (6.7)
DC/NK2	10 (8.5)	3 (8.3)	3 (7.9)	3 (10.0)
D	64 (54.7)	20 (55.6)	23 (60.5)	16 (53.3)
None	14 (12.0)	2 (5.6)	2 (5.3)	5 (16.7)
Other squad	20 (17.1)	9 (25.0)	8 (21.1)	3 (10.0)
Missing	2 (1.7)	-	1 (2.6)	1 (3.3)

**Table 2 sports-07-00148-t002:** Intervention program for short-and long-term intervention (based on the four self-talk (ST) mechanisms postulated by Hardy et al. [1]).

	Main Intervention Question(s)	Aim/Intervention Content
**Preparation**	What is ST? What kind of ST did participants use in the past?	Introduction Learning the possible objectives and effects of systematic ST Participants record ST words and phrases they used in the past Developing an idea of ST
**Behavioral mechanism**	‘How do I optimize movement?’	Understand the conscious influence of ST on movements Developing ST to: - Optimize activation regulation - Improve motor skills - Improve potential movement deficits - Understand the influence of ST on attention and cognition
**Cognitive mechanism**	‘How do I improve my attention?’	Participants record their individual focus during training and competition Developing ST to: - Remain focused on performance detail - Increase attention and avoid distraction - Reduce interfering thoughts
**Affective mechanism**	‘How do I regulate my emotions?’	Understand the role of ST to regulate emotional reactions Participants reflect their positive and negative emotions during training and competitions Integration of systematic relaxation methods (breath control, progressive muscle relaxation) Developing ST to: - Regulate emotional reactions - Regulate activation - Control affects - Decrease dysfunctional states (e.g. state anxiety)
**Motivational mechanism**	‘How do I modify my motivation?’	Understand the encouraging role of ST on motivation Developing ST to: - Improve and maintain motivation - Optimize self-appraisal (e.g. self-efficacy)

**Table 3 sports-07-00148-t003:** Means and standard deviations for three measurement times, and results of 3 (Time) × 3 (Group) multi- and univariate analyses of variance for state anxiety, trait anxiety, volitional skills, self-efficacy, and coach-rated performance.

					Control Group			Short-Term Intervention			Long-Term Intervention		
					Time 1	Time 2	Time 3	Time 1	Time 2	Time 3	Time 1	Time 2	Time 3
Variable	(m)*F* Time (*df*)	*p*; *ƞ*^2^	(m)*F* T × G (*df*)	*p*; *ƞ*^2^	*M_C_* *SD_C_*			*M_STI_* *SD_STI_*			*M_LTI_* *SD_LTI_*		
**State anxiety: WAI-S**	1.79 (6,304)	0.10; 0.03	2.26 (12,402)	0.01 0.16									
Somatic state anxiety	1.48 (2,154)	0.23; 0.02	2.49 (4,154)	0.05 0.08	5.92_a_ 2.41	6.80_b_ 2.68	6.24_a_ 2.62	6.59_a_ 2.52	5.85_b_ 2.49	5.66_c_ 1.67	7.12_a_ 2.66	6.15_b_ 1.99	6.36_b_ 2.59
Cognitive state anxiety	1.49 (2,154)	0.23; 0.02	1.65 (4,154)	0.17 0.02	7.40 2.87	7.84 3.00	8.01 3.55	8.07 2.67	7.30 2.69	8.13 3.74	8.68 3.21	7.39 2.82	7.64 3.01
State self-confidence	3.33 (2,154)	0.04; 0.04	3.10 (4,154)	0.02 0.08	10.72_a_ 2.72	11.19_b_ 2.18	9.96_a_ 2.49	9.85_a_ 2.58	10.41_a_ 3.28	10.30_a_ 3.24	9.96_a_ 2.65	10.96_b_ 2.71	11.36_c_ 2.93
**Trait Anxiety: WAI-T**	5.23 (6,304)	<0.001; 0.09	1.33 (12,402)	0.20 0.03									
Somatic trait anxiety	10.66 (2,154)	<0.001; 0.12	0.85 (4,154)	0.50 0.02	10.32 3.76	9.20 3.58	8.60 3.40	9.84 3.41	9.00 3.74	9.31 3.33	9.84 2.81	8.73 3.63	8.57 3.79
Worry	7.84 (2,154)	<0.001; 0.09	1.31 (4,154)	0.27 0.03	10.04 3.13	9.58 2.97	9.08 3.46	9.19 2.95	8.37 3.70	8.82 3.43	9.99 3.38	8.57 3.22	8.32 3.42
Concentration disruption	6.86 (2,154)	<0.001; 0.08	1.63 (4,154)	0.17 0.04	6.04 1.59	5.80 1.44	6.16 1.99	7.48 2.99	6.52 2.69	6.42 2.47	6.61 2.04	5.79 1.93	5.89 1.91
**Volitional skills (VQS)**	1.39 (8,70)	0.22; 0.14	2.25 (16,140)	0.006 0.20									
Self-optimization	2.32 (2,154)	0.008; 0.03	6.97 (4,154)	<0.001 0.15	57.88_a_ 11.33	55.32_b_ 11.53	51.66_c_ 11.12	57.22_a_ 13.21	60.50_a_ 13.32	59.95_a_ 13.63	55.58_a_ 15.09	61.56_b_ 17.33	62.40_c_ 16.97
Self-impediment	2.05 (2,154)	0.75; 0.01	1.79 (4,154)	0.13 0.04	10.80 3.87	11.80 4.60	12.52 4.74	13.87 5.61	12.61 5.89	13.13 5.33	12.88 4.48	12.27 5.01	12.58 4.13
Energy deficit	1.37 (2,154)	0.92; 0.03	1.98 (4,154)	0.10 0.05	9.05 5.63	10.07 6.31	12.18 7.91	10.54 8.47	9.85 6.98	9.18 7.71	9.32 6.93	9.98 7.20	11.60 8.35
Loss of focus	2,154 (2,154)	0.06; 0.02	0.83 (4,154)	0.50 0.02	8.08 3.44	8.08 5.50	8.96 6.33	10.52 6.39	9.47 6.67	9.41 5.78	9.28 5.34	7.80 5.49	8.37 5.87
**Self-efficacy**	11.18 (2,76)	<0.001; 0.13	4.20 (4,152)	0.002 0.10	27.17_a_ 4.74	27.72_a_ 4.82	26.87_a_ 4.93	27.89_a_ 3.42	29.81_b_ 4.86	29.05_a_ 4.89	27.70_a_ 4.53	30.82_b_ 4.78	31.50_b_ 5.06
**Coach-rated perfor** **mance**	10.66 (2,75)	<0.001; 0.22	3.47 (4,150)	0.01 0.08	25.33_a_ 5.41	26.17_a_ 5.65	27.52_a_ 5.04	27.09_a_ 3.96	27.54_a_ 4.86	28.00_a_ 6.09	25.96_a_ 3.92	28.82_b_ 3.09	27.82_b_ 4.42

*Notes*. T × G = Time × Group interaction effect; *M*, *SD* = means and standard deviations; subscripts in the headline: *C* = control group, *STI* = short-term intervention group, LTI = long-term intervention group; in case of interaction effects, subscript: identical letters identify significant differences between measurement times per group based on Bonferroni post-hoc tests (*p* < 0.05).

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
