# Peer review of "Effects of Self-Talk Training on Competitive Anxiety, Self-Efficacy, Volitional Skills, and Performance: An Intervention Study with Junior Sub-Elite Athletes"

_sports, 2019, doi:10.3390/sports7060148_

Reviewer 1 Report

This randomized controlled trial aimed to investigate the impact of self-talk on outcomes of junior elite athletes.

Major limitations/recommendations:

1) Abstract: Please clarify and modify ‘results’ and ‘conclusions’.

2) The ‘introduction’ is a bit long and difficult to follow. Therefore, it would be necessary to simplify.

3) A rationale for the exclusion/inclussion criteria is necessary.

4) Please add “data analysis plan”

5) Please provide the code that was assigned for the university’s ethics committee to your project.

6) Please state whether you follow the declaration of Ethics (Helsinki) Committee.

7) I would like to know if groups are similar or not regarding sociodemographic characteristics. In fact, if groups were different, it would be necessary to control these potential confounding variables.

8) Limitations of the study: Absence of a cognitive schemas measurements, personality traits…

9) Limitations of the study: Absence of hormonal markers important for this population (e.g. testosterone, cortisol, DHEA, etc).

10) It would be necessary to apply Bonferroni correction for multiple comparisons.

11) Please add the exact “p” value.

12) I would recommend to modify the discussion in order to moderate your conclusions according to study limitations (e.g. relatively homogeneous sample, absence of biological markers…). Furthermore, it should be necessary to include the applications of your study.

 Author Response

We would like to thank the Reviewers for their helpful and very supportive comments on our study. The Reviewers’ comments are numbered consecutively for every reviewer. Please find below our answers. We marked our changes in the manuscript in red in order for you to identify the changes we made more easily. The pages and lines in the response refer to the revised manuscript. We also followed your suggestion and consulted a native speaker.

 Reviewer #1

 Comments and Suggestions for Authors

This randomized controlled trial aimed to investigate the impact of self-talk on outcomes of junior elite athletes.

 Major limitations/recommendations:

R#1-1) Abstract: Please clarify and modify ´results` and ´conclusions`.

> We specified the results more clearly and the main conclusion accordingly.

R#1-2) The ‘introduction’ is a bit long and difficult to follow. Therefore, it would be necessary to simplify.

> We shortened one paragraph (P: 2, L: 85-90), and included a sentence about the novelty aspects of our study (P: 3, L: 116-120). “Finally, the duration of ST training remains under scrutiny. Therefore, the novelty of our study consists of the sample composition (male and female competitive athletes), of the mixture of trait (gender, trait anxiety, general self-efficacy, volition) and state variables (state anxiety, performance), and of testing differential effects of ST training duration.”

R#1-3) A rationale for the exclusion/inclusion criteria is necessary.

> We added on page: 5, line: 189 “Criteria for including participants were 12 years of age and participation in a competitive sport activity. Both criteria were chosen to capture the thus far underrepresented sample of junior sub-elite athletes.

R#1-4) Please add “data analysis plan”.

> Thank you for the note. We have added a chapter “Data Analysis Plan” P: 11, L: 376

R#1-5) Please provide the code that was assigned for the university’s ethics committee to your project.

> Thank you for this and the next note. We have added the code for the university’s ethics committee (P: 7, L: 215-216).

R#1-6) Please state whether you follow the declaration of Ethics (Helsinki) Committee.

> Please see page: 20, from line: 691 “declaration”.

R#1-7) I would like to know if groups are similar or not regarding sociodemographic characteristics. In fact, if groups were different, it would be necessary to control these potential confounding variables.

> We have included another table with socio-demographic details of the sample (Table 1, P: 5, from L: 200) as well as results regarding possible socio-demographic differences between the three groups (P: 11, L: 378) “Univariate analyses of variance and Chi-square test were conducted to examine possible sociodemographic and sports-related differences between the three groups (short-term intervention, long-term intervention, control group).” In this line, we stated on page 11, line 392: “Univariate analyses of variance revealed no significant group differences regarding age (p = .163), years in sport (p = .751), training sessions per week (p = .482), and training hours (p = .760). Furthermore, the Chi-square test revealed no significant differences regarding competition participation (p = .452), competition level (p = .610), or squad category (p = .612).

R#1-8) Limitations of the study: Absence of a cognitive schemas measurements, personality traits...

> We agree with the reviewer, that the assessment of personality traits such as achievement motivation or cognitive schemas measurements would have complemented the data. However, the athletes had to fill out four questionnaires and almost 100 items of dependent variables (including socio-demographic data). To avoid the situation the participants become tired of filling out tests, we restricted the number of questionnaires to the so far underrepresented psychological variables. Nevertheless, competitive trait anxiety (Brand et al.), volition (Kuhl & Fuhrmann) and self-efficacy (Hinz et al.) are understood as personality variables.

R#1-9) Limitations of the study: Absence of hormonal markers important for this population (e.g. testosterone, cortisol, DHEA, etc).

> Thank you for your comment. We agree with the reviewer. The chapter “Limitations” comprises now (P: 18, L: 622): “We also suggest the use of hormonal markers (e.i., testosterone, cortisol, etc) to evaluate the athletes` performance.”

R#1-10) It would be necessary to apply Bonferroni correction for multiple comparisons.

> Change was made. Note in chapter “Data Analysis Plan” page: 11, line: 376 and table 3 (P: 14, L. 429-430).

R#1-11) Please add the exact “p” value.

> Change was made.

R#1-12) I would recommend to modify the discussion in order to moderate your conclusions according to study limitations (e.g. relatively homogeneous sample, absence of biological markers…). Furthermore, it should be necessary to include the applications of your study.

> Thank you for your comment. We followed your suggestion and also included on page 19 from line 637 applications (chapter 7.)

Reviewer 2 Report

Please include in Introduction, more clearly, which is the novelty of this study.

Please detailed how was been recruited the participants from different individual  sports......after what criteria?

Line 200 - What represent ''resp." ???

Please include in chapter results others tables with the results of participants. One table it is not enough!!!!!!!!!!!!!!!!!

Author Response

We would like to thank the Reviewers for their helpful and very supportive comments on our study. The Reviewers’ comments are numbered consecutively for every reviewer. Please find below our answers. We marked our changes in the manuscript in red in order for you to identify the changes we made more easily. The pages and lines in the response refer to the revised manuscript. We also followed your suggestion and consulted a native speaker.

 Reviewer #2

 Comments and Suggestions for Authors:

R#2-1) Please include in Introduction, more clearly, which is the novelty of this study. (N.W.)

> Thank you for your comment. We included a sentence about the novelty aspects of our study (P: 3, L: 116-120): “Finally, the duration of ST training remains under scrutiny. Therefore, the novelty of our study consists of the sample composition (male and female competitive athletes), of the mixture of trait (gender, trait anxiety, general self-efficacy, volition) and state variables (state anxiety, performance), and of testing differential effects of ST training duration.”

R#2-2) Please detailed how was been recruited the participants from different individual sports ......after what criteria?

> Thank you for this note. In the chapter “Design and Procedure(P: 7, L: 216-218) we added the sentences: “Different sport associations and one elite sport school were contacted and informed about the study. In the elite school a project kick-off meeting with teachers and coaches was organized to present the study.”. Furthermore, we added on page: 5, line: 189-191 “Criteria for including participants were 12 years of age and participation in a competitive sport activity. Both criteria were chosen to capture the thus far underrepresented sample of junior sub-elite athletes.

R#2-3) Line 200 - What represent ''resp."???

> “Resp.” can be understood as “more specifically”. We apologize for the confusion and replaced this word using “or” (P: 5, L: 188).

R#2-4) Please include in chapter results others tables with the results of participants. One table it is not enough!!!!!!!!!!!!!!!!!

> We have included another table with socio-demographic details of the sample (Table 1, P: 5, from L: 200) as well as results regarding possible socio-demographic differences between the three groups (P: 11, L: 378) “Univariate analyses of variance and Chi-square test were conducted to examine possible sociodemographic and sports-related differences between the three groups (short-term intervention, long-term intervention, control group).” In this line, we stated on page 11, line 392: “Univariate analyses of variance revealed no significant group differences regarding age (p = .163), years in sport (p = .751), training sessions per week (p = .482), and training hours (p = .760). Furthermore, the Chi-square test revealed no significant differences regarding competition participation (p = .452), competition level (p = .610), or squad category (p = .612).

Reviewer 3 Report

This research article reports new data on the psychological effects of the time (duration) of self-talk on junior elite athletes (N=117, 55 female, 62 male), randomly assigned to two experimental groups and a control group. The experimental groups received self-talk training either for one week, or 18 eight weeks, the control group did not receive self-talk training. Psychological variables of competitive anxiety, volitional skills, self-confidence, and coach performance ratings were assessed three times before and after the test phase. Statistical analyses revealed significant interaction between the effects of duration of self-talk training and the group factor on somatic state anxiety, self-confidence, and performance as rated by the coach. The results are discussed in the light of Hardy's model of interaction between Self-Talk, Competition Anxiety, Motivation, and Performance. The major conclusion drawn is that sustained (in time) Self-Talk training improves both psychological variables and performance outcome in young competitive athletes.

The study appears well-performed, and the conclusions drawn warranted in light of the results and analyses. Some sections of the article require minor revisions of grammar and style; I recommend the authors consult with a native speaker before submitting a final version, or use a language editing service.

Author Response

We would like to thank the Reviewers for their helpful and very supportive comments on our study. The Reviewers’ comments are numbered consecutively for every reviewer. Please find below our answers. We marked our changes in the manuscript in red in order for you to identify the changes we made more easily. The pages and lines in the response refer to the revised manuscript. We also followed your suggestion and consulted a native speaker.

 Reviewer #3

 Comments and Suggestions for Authors:

This research article reports new data on the psychological effects of the time (duration) of self-talk on junior elite athletes (N=117, 55 female, 62 male), randomly assigned to two experimental groups and a control group. The experimental groups received self-talk training either for one week, or 18 eight weeks, the control group did not receive self-talk training. Psychological variables of competitive anxiety, volitional skills, self-confidence, and coach performance ratings were assessed three times before and after the test phase. Statistical analyses revealed significant interaction between the effects of duration of self-talk training and the group factor on somatic state anxiety, self-confidence, and performance as rated by the coach. The results are discussed in the light of Hardy's model of interaction between Self-Talk, Competition Anxiety, Motivation, and Performance. The major conclusion drawn is that sustained (in time) Self-Talk training improves both psychological variables and performance outcome in young competitive athletes.

R#3-1) The study appears well-performed, and the conclusions drawn warranted in light of the results and analyses. Some sections of the article require minor revisions of grammar and style; I recommend the authors consult with a native speaker before submitting a final version, or use a language editing service.

> Thank you for your comment and the positive feedback. We followed your suggestion and consulted a native speaker.

Round  2

Reviewer 1 Report

The authors have made a great effort to improve their article. Therefore, I consider that it is already adequate to be accepted. Moreover, I encourage you to continue researching in this line of research and it would be interesting to include biological variables in your future research.

Reviewer 2 Report

-